# Native Spider Silk-Based Antimicrobial Hydrogels for Biomedical Applications

**DOI:** 10.3390/polym13111796

**Published:** 2021-05-29

**Authors:** Sinith Withanage, Artemii Savin, Valeria Nikolaeva, Aleksandra Kiseleva, Marina Dukhinova, Pavel Krivoshapkin, Elena Krivoshapkina

**Affiliations:** SCAMT Institute, ITMO University, Lomonosova str. 9, 191002 Saint Petersburg, Russia; Withanage@scamt-itmo.ru (S.W.); Savin@scamt-itmo.ru (A.S.); Nikolaeva@scamt-itmo.ru (V.N.); aleksandra_kiseleva@scamt-itmo.ru (A.K.); dukhinova@scamt-itmo.ru (M.D.); krivoshapkin@scamt-itmo.ru (P.K.)

**Keywords:** hyaluronic acid, spider silk, antimicrobial hydrogels, drug delivery, enzymatic degradability, biocompatibility, antimicrobial resistance

## Abstract

Novel antimicrobial natural polymeric hybrid hydrogels based on hyaluronic acid (HA) and spider silk (Ss) were prepared using the chemical crosslinking method. The effects of the component ratios on the hydrogel characteristics were observed parallel to the primary physicochemical characterization of the hydrogels with scanning electron microscopic imaging, Fourier-transform infrared spectroscopy, and contact angle measurements, which confirmed the successful crosslinking, regular porous structure, exact composition, and hydrophilic properties of hyaluronic acid/spider silk-based hydrogels. Further characterizations of the hydrogels were performed with the swelling degree, enzymatic degradability, viscosity, conductivity, and shrinking ability tests. The hyaluronic acid/spider silk-based hydrogels do not show drastic cytotoxicity over human postnatal fibroblasts (HPF). Hydrogels show extraordinary antimicrobial ability on both gram-negative and gram-positive bacteria. These hydrogels could be an excellent alternative that aids in overcoming antimicrobial drug resistance, which is considered to be one of the major global problems in the biomedical industry. Hyaluronic acid/spider silk-based hydrogels are a promising material for collaborated antimicrobial and anti-inflammatory drug delivery systems for external use. The rheological properties of the hydrogels show shear-thinning properties, which suggest that the hydrogels could be applied in 3D printing, such as in the 3D printing of antimicrobial surgical meshes.

## 1. Introduction

Hydrogels are hydrophilic, three-dimensional polymer structures that can absorb amounts of water several times greater than their own weight and size [1,2,3]. During the last decade, hydrogels have progressively evolved, with their characteristics and properties steadily diversifying [4]. Hydrogels can be made from either synthetic or natural polymers. Natural polymer based hydrogels have proven that hydrogels can acquire highly stable physical, chemical, and mechanical properties in their swollen state, which has inspired scientists to develop and use new hydrogels for extraordinary applications [5,6]. The porosity, soft consistency, and ability to absorb biological fluids exhibited by these hydrogels have increased their pote biomedical applications [7,8,9].

Nowadays, scientists use various techniques to produce novel hydrogel materials, with different approaches and techniques aimed at increasing the applicability of desirable characteristics [4,10]. The use of natural biopolymers in hydrogel synthesis has been widely discussed among researchers, which has led to the fabrication of natural polymer-based hydrogels whose natural properties and characteristics have been preserved [11,12,13]. Hydrogels are primarily synthesized via physical and chemical crosslinking of polymers [14,15,16]. Ionic crosslinking, photo crosslinking, and radiation induced crosslinking are some of the more widely used physical crosslinking methods in polymeric hydrogel synthesis [17,18]. Covalent crosslinking, a chemical crosslinking method, is preferable since the resultant polymeric structures are useful for chemical, biological, and physical applications [19,20]. With progressive technological development and the identification of novel aspects of science, hydrogels have begun to play a larger role in major developmental breakthroughs. Hydrogels are currently used in major modern biomedical applications such as drug delivery, cell visualization, hygiene products, 3D printing, etc. [11,21,22,23,24].

Recently, most hydrogels have been fabricated from synthetic polymers since it is more convenient to predesign and predict their behavior and characteristics compared to naturally occurring polymers [25,26]. The extensive involvement of synthetic polymers in different industrial applications affects biodegradability and biocompatibility by producing hazardous outcomes in nature, day to day human activities, and the health of living beings [26,27]. Although researchers were recently able to successfully develop biocompatible and biodegradable hydrogels from synthetic polymers, the deposition of most of these materials is still not eco-friendly since there is no natural origin of these polymers in environmental cycles or activities [28,29,30]. Consequently, these types of materials can interfere with natural cycles and eventually adversely affect biological systems [30,31]. Moreover, the involvement of advanced synthetic polymers affected engagement of natural polymers in current industries, which encloses the extraordinary capabilities of natural polymers over synthetic polymers [32,33]. As a result, researchers have shifted their focus to novel natural polymers, and the characteristics of these materials show extraordinary potential for application in diverse experimental and applicable systems [33,34]. Hence, the research interests on natural polymers largely increased, and currently, there are major industry leaders and institutions involved in innovative investigations involving natural polymers. Natural polymers such as collagen, fibroin, hyaluronic acid, chitosan, and cocoon silk are widely using in hydrogel synthesis and development with the aim of diverse applications [22,35,36,37]. Although naturally occurring, these polymers show concentration-dependent and molecular weight-dependent toxicity according to various acute toxicity studies relevant to different applications [38,39]. 

Spider silk is an extraordinary biopolymer with excellent chemical, physical and biological properties [40,41]. Spider silk also has unique bio-molecular properties compared to other biopolymers such as collagen, polylactic acid, fibrin, alginate, gelatin, etc. Its ability to significantly lower inflammatory responses, as well as act as a scaffold for the growth of different cell types by facilitating cell adhesion, makes spider silk superior to other biopolymers [42,43,44]. Spider silk fiber consists of multiple spider fibrils, which are formed by the beta-sheets combined with amorphous regions. The main amino acids found in spider silk are alanine, glycine, and serine. Alanine provides extraordinary strength, while glycine gives elasticity to the spider silk [41,45]. Recently, spider silk has been used in a wide range of biomaterials, such as films, matrices, and hydrogels, and spider silk-based materials are involved in diverse biomedical applications [46,47,48,49,50]. Recombinant spider silk produced in different host organisms has commonly been used in recent silk-based material synthesis studies with artificial repetitions of known amino acid sequences and predesigned secondary structural advancements. This suggests an extraordinary improvement in the applicability of spider silk [51,52]. 

Nevertheless, the involvement of natural spider silk in the material industry is yet to be developed due to the potential complications and disadvantages of artificial silk synthesis [53]. Mainly, recombinant spider silk synthesis requires extended time and different host organisms to facilitate its production [54]. Different types of recombinant hosts have been investigated recently, and it shows diverse complications of the synthesis process in different hosts [55]. The recombinant production of spider silk has potential disadvantages, such as complicated genetic manipulations with host-dependent generation intervals [55,56]. Furthermore, the genetic rearrangement and the repeated sequence insertions may produce altered or unknown amino acid sequences [57]. The main obstacle, however, is assumed to be the extraction and purification of the spider silk proteins from the host [54]. Different studies confirm that the cytosolic aggregation of silk proteins within the host cytosol may affect the purification and lowering of the protein yield [54]. Based on experience in SCAMT Institute’s insectarium, it was found that one adult spider (based on the example of *Linothele Fallax* spider) can secrete up to 10 mg of silk per day according to different phases of their life cycle, and these spiders can live up to 25 years [58,59]. This amount of spider silk is enough to perform various kinds of experiments. Considering the observations, it is possible to create spider farms and produce the amount of natural silk needed for large-scale industrial applications. Due to its natural origin and sequence, material scientists suggest that natural spider silk may have a wide range of potential applications yet to be discovered.

Hyaluronic acid is one of the most significant components of the extracellular matrix in human skin and is commonly found in epithelial, neural, and connective tissues [60]. Hyaluronic acid contains different chemical groups, mainly carboxylic and hydroxyl, that facilitate crosslinking, resulting in the formation of much more stable polymeric structures [61]. Hyaluronic acid is made of D-glucuronic acid and N-acetyl-D-glucosamine repeats connected with glycosidic bonds to form a glycosaminoglycan structure [62]. In the cosmetic industry, hyaluronic acid plays a significant role as a component of a variety of skincare products, which shows the surprising benefits of maintaining supple skin and promoting healthier skin [63,64]. Hyaluronic acid possesses extensive wound healing ability, and promotes anti-inflammatory and anti-bacterial effects [65,66,67,68,69]. Furthermore, hyaluronic acid stimulates the regeneration of blood vessels in damaged tissues [70]. Hyaluronic acid has been shown to be an excellent material for pain relief in tissue damage, and it can also be applied as a lubricant for bone joints to relieve pain [71,72]. Combining hyaluronic acid with chondroitin sulfate can extensively reduce reflux symptoms when combined with acid reflux medications [73]. In urinary frequency and bladder pain, hyaluronic acid provides pain relief and reduction of urinary frequency symptoms after the infusion of hyaluronic acid into the bladder [74]. 

In recent studies, hyaluronic acid has gained much attention due to its wide applicability. Hyaluronic acid has been involved in a variety of polymer synthesis procedures, leading to the production of different materials, such as hydrogels, with diverse applications mainly in the biomedical, cosmetic, and clinical fields [75]. Hyaluronic acid combined with different biopolymers, such as collagen and chitosan, has recently been investigated. Results showed that hyaluronic acid-based hybrid materials will likely make promising advancements in future research aspects over a diverse scope [76,77]. 

Modification of hyaluronic acid carboxylic groups, with 1-ethyl-3-(3-dimethyl aminopropyl)-carbodiimide (EDC)/N-Hydroxysuccinimide (NHS) mediated coupling, has been involved in different hydrogel preparation methods [78]. EDC/NHS coupling is used to combine carboxylic groups with amine groups leading to the formation of covalent cross-links between polymers. This crosslinking method has been used to cross-link polymers in recent polymerization techniques. Thus, the involvement of biopolymers in the hydrogel industry expresses futuristic aspects of applications in a vast range of applications in clinical and biomedical settings. The combination and synthesis of hybrid polymeric systems with the combined physical, chemical, and biological characteristics of biopolymers seem to be widening current research scopes and innovative industrial applicability.

In this research, hyaluronic acid and natural spider silk protein-based hydrogels were prepared using EDC/NHS crosslinking. The study utilized novel aspects of hyaluronic acid and natural spider silk-based hydrogel formulation and their involvement in modern biomedical applications. The main priority of the initial formulation is to combine the unique properties of the natural spider silk proteins with the properties of hyaluronic acid. Physicochemical and biological characterization of the novel hydrogel materials were initially identified to evaluate the applicability of the hydrogels to overcome current global problems in the biomedical industry. Antimicrobial resistance is one of the main global problems in the biomedical industry. Drug delivery systems and drug carriers are still to be developed due to their immunogenicity and toxicity. In clinical settings, surgical infections are more common due to the lack of antimicrobial surgical materials. The subjected hydrogel formulations suggest promising capabilities with regard to the biomedical industry as an external drug carrier, microbial inhibitory material, 3D printing material, and optimum material for electrophoretic drug delivery systems.

## 2. Materials and Methods

### 2.1. Materials

Spider silk (ITMO University spider insectarium), trifluoroacetic acid (TFA) (ReagentPlus^®^, 99% by Sigma Aldrich, St. Louis, MO, USA), enzyme trypsin (BRP, European Pharmacopoeia (EP) Reference Standard by Sigma Aldrich), enzyme chymosin (Chemsavers CAS: 9001-98-3, >20 Units/mg), ibuprofen sodium salt (analytical standard by Sigma Aldrich, ≥98% (GC)), *E. coli* (Nova Blue strain, TcR, AmpR, Novagen, Inc., Madison, WI, USA) and *Micrococcus* bacteria (Collection of the Pasteur Research Institute of Epidemiology and Microbiology), EDC (1-ethyl-3-(3-dimethyl aminopropyl)-carbodiimide) (pure > 98% AT by Sigma Aldrich, St. Louis, MO, USA), NHS (N-Hydroxysuccinimide) (assay: 98%, MW:115.09 g/mol by Sigma Aldrich, St. Louis, MO, USA), phosphate-buffered saline (PBS) (Gibco^®^, 1 X, pH = 7.4), hyaluronic acid sodium salt (*M*_W_: 750,000 by Sigma Aldrich), MES (4-(N-morpholino)ethanesulfonic acid) (BioUltra, ≥99.5% (T), Sigma Aldrich, St. Louis, MO, USA), 0.1 M HCl, 70% ethanol, CH_3_COONa (for molecular biology, ≥99% by Sigma Aldrich, St. Louis, MO, USA), sodium chloride (NaCl) (ACS reagent, ≥99.0%, Sigma Aldrich), agar (Microbiology grade), 3-(4,5-dimethylthiazol-2-yl)-2,5-diphenyltetrazolium bromide (MTT) (Sigma-Aldrich, 98%), dimethyl sulfoxide (DMSO) (VWR chemicals, Molecular Biology Grade), penicillin streptomycin (Biolot, Russia), and phosphate buffer solution tablets (VWR) were used in our research. Milli-Q grade deionized water (DI, 15 MΩ cm resistivity) was used in all experiments.

### 2.2. Preparation of Spider Silk Solution

Spider silk (without impurities) was obtained from the SCAMT Institute insectarium, and the fibers were cut into approximately 1–2 mm pieces. Trifluoroacetic acid (TFA) was used to dissolve the spider silk—20 mg of dried silk fibers were dispersed in 1 mL of TFA solvent, under constant mixing at 500 rpm at room temperature. Samples were examined each 15 min for complete solubilization. TFA appeared to dissolve silk instantaneously, minimizing experimental time. The fluorinated spider silk solution (18 mg/mL) was carefully transported into a dialysis membrane (Zellu Trans Dialysis Tube T4, Scienova GmbH, pore diameter 12–14 kDa). Further, the sample was fixed on a laboratory tripod and placed into an external chamber of the buffer solution. The composition of the buffer system was as follows: 400 mL of DI water, 41.05 g of CH_3_COONa (500 mMol), and 11.7 g of NaCl (200 mMol).

The dialysis was performed for 5 days at room temperature with gentle stirring of the buffer system. The constant identification of the pH was performed by pH meter ST3100-B (OHAUS, Moscow, Russia), and evaluation of the presence of fluoride ions was performed using CaCl_2_ quality reactions. The pH values of the external chamber and inside the dialysis membrane were observed every 6 h. The buffer system was refreshed every 2 h during the first two days of the experiment and left overnight. The buffer refreshment was performed every 6 h after the second day until the end of the dialysis on the fifth day. The final pH of the silk protein solutions was 7. Figure 1 illustrates the preparation procedure of the spider silk protein solution schematically.

### 2.3. Preparation of Hyaluronic Acid/Spider Silk-Based Hydrogels

Hyaluronic acid/spider silk (HA/Ss)-based hydrogels were prepared via EDC/NHS mediated polymerization. A 0.1 mol/L MES (4-(N-morpholino)ethanesulfonic acid) solution was prepared by dissolving in deionized water (pH = 4.5). MES is suggested to be the best activation buffer for EDC/NHS reactions, optimizing the exact bonding mechanism between carboxylic groups and amines [79]. Hyaluronic acid was dissolved in 0.1 mol/L MES solution for 6 h under magnetic stirring, according to the different preparation ratios as shown in Table 1, until a homogenous hyaluronic acid solution was obtained. Spider silk solution was then added to the hyaluronic acid homogenous solution and mixed under magnetic stirring for 3 h to obtain HA/Ss mixtures of different ratios, as shown in Table 1. HA/Ss mixture treated with NHS (100 mM) followed by drop-wise adding of EDC (100 mM) with constant magnetic stirring. Hyaluronic acid/spider silk (HA/Ss) based hydrogels were prepared within 10–20 min of constant stirring. The resulting hydrogels were initially washed with distilled water and immersed in 75% ethanol for 30 min to remove the residual monomers. Finally, the hydrogels were washed with PBS (pH = 7.4) prior to further use. Washed PBS was collected and the pH changes were observed once the residual monomers were eliminated. The hydrogels were washed until the pH remained constant compared to the pH of the initial PBS before washing.

### 2.4. Freeze Drying of Hydrogels

Hydrogels were obtained and dried under low temperatures (−50 °C) at 0 atm pressure in the freeze dryer. All the water content occupied in the hydrogels was removed under dry freezing. After 24 h of dry freezing, samples were removed from the dry freezer and prepared for further experiments. In this experiment, we used a hydrogel with an initial weight of nearly 1 g and observed the dry weight in order to measure the swelling ability.

### 2.5. Physical and Chemical Characterization of Hydrogels using Scanning Electron Microscopy

Samples obtained after 24 h of freeze-drying followed by three days of 95% ethanol immersion were carefully cut into small pieces and prepared for scanning electron microscopy (SEM) images. Ethanol immersion facilitates the stability of the porous structures, leading to obtaining optimum SEM images. SEM images were obtained using Oxford Scanning Electron Microscope TESCAN Vega 3.

### 2.6. Fourier-Transform Infrared Spectroscopy

Fourier-transform infrared (FTIR) spectroscopic measurements were taken, leading to the characterization of the functional groups and the chemical composition of the hydrogels. The hydrogels were initially freeze-dried. Spectra were acquired with a Nicolet iS10 FTIR spectrometer (Thermo Fisher Scientific). The spectra were measured in the range from 4000–400 cm^−1^ in absorption mode. Dry samples were treated with crystalline KBr (1 mg of sample:199 mg of KBr) and then pressed into a disk. The data were obtained at a resolution of 0.5 cm^−1^ with 40 cumulated scans and a signal-to-noise ratio of 40,000:1. The fitting of the amide I peak was performed using Gaussian functions. The criteria employed for the determination of the initial conditions of the fitting were based on the second derivative of the experimental spectrum. The minima of the second derivative were used to determine the number and position of the Gaussian functions used for the fitting. The initial full width at half maximum (FWHM) was initially fixed to 8 cm^−1^ for every Gaussian used in the fitting of the spectrum. Finally, the fitting was performed after applying a linear baseline. The entire fitting process was carried out with Omnic 9 software. The areas of the Gaussians were used to estimate the content of the different secondary structures according to the band assignments [80]. 

### 2.7. Contact Angle Measurements of Hyaluronic Acid /Spider Silk-Based Hydrogels

The contact angle measurements were performed with the drop shape analyzer by KRUSS DSA25 equipped with a video-capturing system. The wettability and the hydrophilic capabilities of the hydrogels were determined by the contact angle measurements.

### 2.8. Rheological Properties of Hyaluronic acid /Spider Silk-Based Hydrogels

Viscosity and rheological properties were observed using DHR-1 rotational rheometer (Discovery Hybrid Rheometers, USA). Initially, 1ml of hydrogel sample was placed on the stage plate of the rheometer, and the rotor distance was adjusted to 0.5 cm. The measurements were taken in the range of 0.1 s^−1^ to 30 s^−1^ of shear rate. Viscosity measurements were taken under 22 °C and 37 °C, and pH = 4.5 (crosslinking pH) and pH = 7.2 (physiological pH) conditions. Viscosity and stress of the hydrogels were graphed over shear rate. The viscosity can be described using shear stress and shear rate as follows [81];
(γ) = dγ/dt
(σ) = F/A
(τ) = D/H
where γ represents shear rate (s^−1^), σ is shear stress, dγ is change in strain, dt is change in time, F represents force, A represents the area, D is deformation, H is height, and τ represents shear strain. Finally, the viscosity can be mathematically presented as follows;
ή = σ/γ
where ή is viscosity, σ is shear stress, and γ is shear rate.

### 2.9. Swelling Ability of Hyaluronic Acid/Spider Silk-Based Hydrogels in Water

The swelling ability of hydrogel samples was obtained using swelling weight over time, and the swelling degree was calculated according to the following equation [82]. 1 g of hydrogels were soaked in water and the swelling weight at different time intervals was observed until attaining complete maximal equilibrium under 22 °C. Five repetitions of the procedure were performed with each hydrogel preparation leading to a calculation of the standard error with the identification of standard deviation.
$$Q=\left(W_{0}-W_{d}\right)/W_{d}$$

Q—Swelling degree

W_0_—Wet weight of hydrogels

W_d_—Dry weight of the hydrogels

### 2.10. Gel Fraction and Crosslinking Density

The hydrogels were obtained after three-step washing (as described in Section 2.3) to remove residual monomers. The dried hydrogels were exposed to distilled water and left to swell until the system reached maximum equilibrium state. The maximally swollen hydrogels were initially weighed and left uncovered at room temperature for 3 days. The samples were further dried in an oven for 3 h under 70 °C before measuring the final weight. The solution fraction and the gel fraction of the hydrogels were calculated by their insoluble part after extraction as follows [83];
GF(%) = [Mf/Mi] × 100%
where GF represents gel fraction, Mf is the weight of the dried sample, and Mi represents the weight of the original hydrogel before drying.

The equilibrium volume fraction of the hydrogels were calculated using the following equation [84];
Vp = 1/Q
where Vp is equilibrium volume fraction and Q represents the swelling ratio.

Crosslinking density of the hydrogels was calculated by the Flory–Rehner equation, which is based on equilibrium swelling of three-dimensional polymers. The Flory–Rehner equation can be described as follows [84,85];
$$v=\frac{-\ln\left(1-\mathrm{Vp}\right)+\mathrm{Vp}+{\mathrm{XVp}}^{2}}{V1\left({\mathrm{Vp}}^{\frac{1}{3}}-\frac{\mathrm{Vp}}{2}\right)}$$
where v is crosslinking density, Vp is equilibrium volume fraction at swelling state, and X represents the solvent interaction Flory parameter.

### 2.11. Shrinking Ability of the Hyaluronic Acid/Spider Silk-Based Hydrogels

The shrinking ability of the hydrogels was measured at 37 °C and 1 atm [86]. Hydrogels were left uncovered and the weight loss was observed over time. The weight loss is subjected to the loss of water volume incorporated within hydrogels in physiological conditions by evaporation. The standard illustration of the results was obtained by repetitions of the procedure leading to the identification of the standard error.

### 2.12. Drug Loading in Hyaluronic Acid/Spider Silk-Based Hydrogels

Ibuprofen solution with a concentration of 4 mg/mL prepared by dissolving ibuprofen in water and 0.1 g of hydrogels were soaked in the known amount of solution, and the absorbed drug amount (drug loading capacity and drug entrapment efficiency) was calculated by obtaining the residual drug amount in the remaining medium, followed by the absorption system reached to its swelling equilibrium. The remaining drug concentration was measured by UV-Vis Spectroscopy (Agilent Cary 8454 UV–Visible spectrophotometer) with obtaining values according to the ibuprofen calibration curve. The drug loading capacity of the hydrogels is calculated considering the maximum drug absorption in the Ibuprofen swelling equilibrium state. Drug entrapment efficiency and drug loading percentage was obtained using the following equations [87];
EE (%) = [(DA − FUD)/DA] × 100%
DLC (%) = [ED/EHW] × 100%
where EE—entrapment efficiency; DA—drug added; FUD—free unentrapped drug; DLC—drug loading capacity; ED—entrapped drug; EHW—entrapped hydrogel weight.

### 2.13. Drug Release Studies of Hyaluronic Acid/Spider Silk-Based Hydrogels

Drug release studies were observed in different sample preparations, and the drug release rate over time was graphed. The drug delivery system of ibuprofen works according to the concentration gradient diffusion. Drug loaded hydrogels consist of a high concentration of ibuprofen, and the drug particles diffuse against phosphate buffer solution in the drug release medium. Drug-loaded hydrogels were soaked in PBS solution (pH = 7.4). The drug release was observed under 37 °C by collecting the PBS in different time intervals and immediately replacing the PBS in the delivery chamber. The collected PBS with released drug content was analyzed by UV-Vis Spectroscopy (Agilent Cary 8454 UV–Visible spectrophotometer) and the drug release concentration at different time intervals was obtained. The absorbance spectra of the ibuprofen were measured with 10 mm cell against blank solvent. Ibuprofen shows a well-defined maximum absorption spectra (λmax) at 221 nm. Considering the λ_max_ of ibuprofen, absorption of the released drug was measured at 221 nm wavelength, and the calibration curve provided the exact concentrations of drugs relevant to absorbance readings.

### 2.14. Conductivity of Hyaluronic Acid/Spider Silk-Based Hydrogels

The conductivity of the hydrogel samples was obtained usinga two probe digital multi-meter (MASTECH MAS830L). The multi-meter was fixed vertically, and the two probes of the multi-meter were dipped into the hydrogel samples, maintaining a distance of 10 mm. The resistance of the hydrogels was obtained by the readings of the multi-meter and the conductivity was calculated according to the following equation at 22 °C [88]. 

Conductivity can be defined as:G = 1/R

G—Conductivity

R—Resistivity

### 2.15. Biodegradability Studies of Hyaluronic Acid/Spider Silk-Based Hydrogels

Biodegradability studies of the hydrogels were observed using the enzymes trypsin and chymosin. Approximately 1 g of each swollen hydrogel sample was treated with 1ml of 1% enzyme (Trypsin/Chymosin) and left to incubate at 37 °C. The weights of degradation were measured by removing the residual minor polymeric particles at different time intervals while preserving the main hydrogel mass. The enzyme replacement was performed once per day [89]. The sample characteristics were observed over time, leading to the identification of the degradability of hydrogels with chymosin and trypsin enzymes. Five repetitions of the procedure were performed, leading to the obtainment of the standard error with the identification of standard deviation.

### 2.16. Antimicrobial Ability of Hyaluronic Acid/Spider Silk-Based Hydrogels

The antibacterial activity of the hydrogels was tested on two bacteria species–*Escherichia coli* (Nova Blue strain, TcR, AmpR, Novagen, Inc., Madison, WI, USA) and *Micrococcus sulfuricum*—from the collection at the Pasteur Research Institute of Epidemiology and Microbiology. The antibacterial ability of the samples was observed using the inhibitory band microbiological test. In this experiment, *Escherichia Coli* Nissle was used as the model to represent gram-negative bacteria, and *Micrococcus sulfuricum* was used as the model to represent gram-positive bacteria to observe the inhibitory effects of the hydrogel samples. *E. coli* and *Micrococcus sulfuricum* were prepared separately in a lysogenic broth (LB) solution and incubated overnight while stirring at 37 °C. The prepared bacterial samples were dissolved in 0.9% NaCl and applied to agar culture plates. Hydrogel samples were carefully placed on the spread of bacteria on culture plates. Covered plates were incubated for 18 h at 37 °C and the inhibition bands were observed. In this experiment, three repetitions were done with both bacteria.

### 2.17. Cytotoxicity of Hyaluronic Acid/ Spider Silk-Based Hydrogels

Human postnatal fibroblasts (HPF) cells were used for the cell proliferation and cytotoxicity experiments. The cells were cultured in Dulbecco modified Eagle’s medium (DMEM) supplemented with 10% fetal bovine serum and gentamicin (50 µg/mL) at 37 °C and a 5% CO_2_ atmosphere. Initially, hyaluronic acid/spider silk-based hydrogels were immersed in ethanol for 30 min and washed with PBS (pH = 7.4) three times. The hydrogels were exposed to ultraviolet light for 30 min before being used in the cell experiments. The prepared cells were cultivated on the hydrogel surfaces and kept undisturbed at 37 °C and 5% CO_2_ atmosphere. The HPF cell proliferation on the hydrogel surfaces was observed under the light microscope over 3 days of duration.

The cell culture media for HPF were Dulbecco modified Eagle’s medium (DMEM) supplemented with 10% fetal bovine serum, penicillin, and streptomycin at concentrations of 50 U/mL and 50 µg/mL, respectively, at 37 °C and a 5% CO_2_ atmosphere. The day before testing, we filtered spider silk protein through a 0.22 µm filter (JET Biofil) and added penicillin and streptomycin solution at the same concentration as for the media in the stock solution of filtered spider silk protein and HA. Cell viability studies were performed by MTT assay. For that, all cells (1 × 10^4^ cells per well) were seeded on a 96-well plate (Eppendorf). After overnight incubation, HA (0.06:15 mg/mL) spider silk (0.0125:3.2 mg/mL), HA/spider silk gel (15 mg/mL and 3 mg/mL, respectively) were pipetted in the wells. To account for the absorption of the gel itself for each sample and volume of gel which does not provide cells with nutrients as well as media. Additional background controls and DMEM + PBS were added during the experiment. Then, the plate was incubated for 72 h. After incubation, the MTT solution at a concentration of 0.5 mg/mL was added for 2 h and then aspirated. Subsequently, DMSO was added to dilute the formed formazan and the absorbance was measured at 570 nm on a Spark^®^ Multimode Microplate Reader by Tecan. Cytotoxicity was calculated as the percentage ratio of absorbance in the wells with solution-treated cells in them to that of control cells (100%).

## 3. Results and Discussion

### 3.1. Preparation of Hyaluronic Acid/Spider Silk-Based Hydrogels

Figure 2 shows the scheme of the EDC/NHS mediated chemical crosslinking method of the hyaluronic acid and spider silk leading to obtaining hyaluronic acid/spider silk-based hydrogels. Hyaluronic acid/spider silk-based hydrogels were prepared using EDC/NHS coupling to crosslink hyaluronic acid with spider silk fibers. The carboxylic groups of hyaluronic acid combined with the amine groups of spider silk proteins. The carboxylic groups of Hyaluronic acid initially react with EDC to form O-acylisourea intermediates in the medium of pH 4.5–5.5. The reaction occurs instantly after the addition of EDC. O-acylisourea intermediate converts directly into amine reactive NHS ester by dissociating EDC and reacting with NHS. In a medium where the pH is maintained at 4.5, the amine groups of dissolved spider silk produce covalent bonds with the carboxyl group of hyaluronic acid by dissociating NHS from the amine reactive NHS esters. EDC and NHS facilitate extraordinary reactivity of carboxylic groups of polymers leading to the carboxylic-amine reactions involved in crosslinking chemistry.

### 3.2. Scanning Electron Microscopic Images of Hyaluronic Acid/Spider Silk-Based Hydrogels

HA/Ss 2 (Table 1) hydrogels were used as the characteristic sample for the illustrated results as shown in Figure 3. The characteristic sample was used due to its standard properties shown in the comparative analysis of physical, chemical, and biological properties, such as biocompatibility, enzymatic degradability, viscosity, etc. The imaging of the freeze-dried hydrogels using scanning electron microscopy revealed the fibrillar structure, and the cross linked fibrillar network confirmed the successful chemical crosslinking between hyaluronic acid and spider silk, leading to the formation of the hydrogel networks. The SEM images under 20 µm and 5 µm of magnification are shown in Figure 3a allowing us to visualize the exact nature of the hydrogels.

### 3.3. Swelling Degree of Hyaluronic Acid/Spider Silk-Based Hydrogels

The swelling degree of hyaluronic acid/spider silk-based hydrogels was investigated, and is shown in Figure 3b for the HA/Ss 2 composition (Table 1). For the hydrogels with a higher hyaluronic acid ratio, higher swelling ability was observed. Furthermore, the increment of the spider silk in the medium shows a slight increase in the swelling ability. (Appendix A) However, it does not show a drastic increment compared to the samples with the higher hyaluronic acid ratio in the hydrogel medium. The maximum swelling degree range of the hyaluronic acid/spider silk-based hydrogels is between 18–26 (*W*/*W*). The results of the swelling degree of hyaluronic acid/spider silk-based hydrogels show excellent applicability in various industries such as drug loading, drug delivery, hygiene products, cosmetics etc.

### 3.4. Gel Fraction and Crosslinking Density of Hyaluronic Acid/Spider Silk-Based Hydrogels

Gel fraction and the crosslinking density of the hydrogels are calculated as described in the experimental section. The gel fraction parameters and the crosslinking density of hydrogels are shown in Table 2.

The gel fraction of the hydrogels is comparable with the component ratios of the hydrogels. The gel fraction was slightly increased with the hyaluronic component ratio increases in the hydrogel preparations. Meanwhile, the higher spider silk concentrations in the hydrogels, the gel fraction, and crosslinking density were all increased. The effects of the hyaluronic acid on the crosslinking density were affected by the presence of spider silk in the preparations. Even when the hyaluronic acid proportion increased in the hydrogels, the crosslinking density was not directly affected provided that the spider silk proportions remain unchanged. Spider silk protein concentrations in the hydrogels directly affected the crosslinking density and structural properties of the hydrogels.

### 3.5. Shrinking Ability of Hyaluronic Acid/Spider Silk-Based Hydrogels

The shrinkage of hyaluronic acid/spider silk-based hydrogels was observed under physiological conditions (37 °C). Depending on the composition of the hyaluronic acid and spider silk in the hydrogels, the shrinking ability fluctuates (Appendix A). Hydrogels with higher ratios of hyaluronic acid and spider silk show less shrinking ability, since the expulsion of water from the structures with high crosslinking density is comparatively low. The maximum shrinkage shows HA/Ss 1 from the hyaluronic acid/spider silk hydrogels. The shrinking ability of hyaluronic acid/spider silk hydrogels ranges from 62–81% of their initial weight. The shrinkage of hydrogels in physiological conditions (37 °C) is shown in Figure 3c for HA/Ss2 (Table 1). The shrinking of hyaluronic acid/spider silk-based hydrogels suggests that their supportive properties can be employed as a delivery material under physiological conditions.

### 3.6. Fourier-Transform Infrared Spectroscopy

Dried hyaluronic acid/spider silk-based hydrogel was processed into a solid, powdery condition and analyzed via an infrared spectroscopy (FTIR) method to characterize the functional groups and the chemical composition of the hydrogels. The peaks analysis of the obtained spectrum revealed characteristic peaks of spider silk peptides in the regions of 1650 cm^−1^, which corresponds to the C = O of the amide group (amide I), and at 1550–1510 cm^−1^, corresponding to the anti-symmetric vibrations of carboxyl groups and the N–H bond (amide II). Peaks at 1230 cm^−1^ are related to mixed C–N stretching and N–H bending vibrations of spider silk amino acids, while a peak at about 3300 cm^−1^ correlated with stretching vibrations of the N–H bond. The characteristic bands corresponding to the vibrations of hyaluronic acid groups were identified as well: a peak at 1030 cm^−1^ relates to vibrations of the ether bond; peaks in the region of 3040–2853 cm^−1^ show a series of bands of (CH)_n_ groups inherent to hyaluronic acid and spider silk. The obtained spectrum for HA/Ss 2 confirms that the main organic components of the gel are hyaluronic acid and spider silk peptides (Figure 3d). The insert on Figure 3d shows the analysis of the secondary structure of the proteins in the silk according to a method described by R. Madurga, A. M. Gañán-Calvo, G. R. Plaza, G. V. Guinea, M. Elices, and J. Pérez-Rigueiro in Green Chem. 2017, 19, 3380 [80]. The results confirm the predominant composition (62%) of beta sheets, 18% of beta turns, 11% of aggregated strands, and a minority (9%) of random coils.

### 3.7. Contact Angle Measurements and FLIP Test of Hyaluronic Acid/Spider Silk Hydrogels

The flip test was performed with the hydrogels under physiological temperature (37 °C) and room temperature (22 °C), leading to the identification of the hydrogels’ stability. The hydrogels show excellent stability under both conditions. The results suggest that spider silk serves as a scaffold for hyaluronic acid/spider silk-based hydrogels, leading to the maintenance of the mechanical properties and the stability of hydrogels. (Insert on Figure 3e). The contact angle measurements of the hyaluronic acid / spider silk-based hydrogels confirmed the hydrophilic properties, one of the major characteristic aspects of the hydrogels. Hyaluronic acid/spider silk hydrogels (HA/Ss 2) show contact angles of 82.7° and 88.3°. Contact angles between 0–90° prove the hydrophilic capabilities of the testing material. The results confirm the potential wettability of hyaluronic acid/spider silk-based hydrogels as an absorbent material with different applications (Figure 3e).

### 3.8. Rheological Properties of Hyaluronic Acid/Spider Silk Hydrogels

The rheological properties of hyaluronic acid/spider silk-based hydrogels were illustrated by the shear rate and shear stress measures for the demonstration of viscosity of the hydrogels. The viscosity of the hyaluronic acid/spider silk-based hydrogels at pH 4.5 (crosslinking conditions) and 22 °C shows an increased viscosity, while the hyaluronic acid/spider silk-based hydrogels at pH 7.2 (physiological medium) and 22 °C shows the lowest viscosity behavior. The viscosity of the hydrogels under physiological temperature (37 °C) and physiological pH (pH 7.2) shows intermediate viscous behavior. The viscosity of the hyaluronic acid/spider silk hydrogels increases once the pH approaches its crosslinking pH, just as the viscosity increases once the temperature approaches physiological temperature. The illustrated curve of viscosity and shear stress over the shear rate suggests that the hydrogels display the typical characteristics of shear thinning. Shear-thinning hydrogels produce a reduction in viscosity upon the application of shear, which can be activated by reversible cross-linking mechanisms. The shear thinning hydrogels are optimal for use as matrices in 3D printing due to their flexibility. 3D printing is one of the modern and novel aspects of science, which is currently enabling extraordinary advancements in science. There are various types of materials that have recently been used for 3D printing for various applications, such as printing of tissue constructs or surgical materials. The 3D printing methods that are currently under development are opening the aspects of further printing of organs or even organisms in the form of “bioprinting”. The viscosity of the hydrogels for biomedical applications is extremely important in determining their applicability. Figure 3f shows the rheological properties of HA/Ss 2 hydrogel.

### 3.9. Drug Loading of Hyaluronic Acid/Spider Silk Hydrogels

Drug loading studies of hyaluronic acid/spider silk-based hydrogels were performed with the anti-inflammatory drug ibuprofen. Ibuprofen is one of the main anti-inflammatory drugs used in current clinical applications [90,91]. The ibuprofen loading into the crosslinked hyaluronic acid/spider silk-based hydrogels’ porous structures was conducted at 24 °C and 1 atm pressure. The drug loading followed the concentration gradient swelling until the system acquired the diffusion facilitated swelling equilibrium. Figure 4a shows the scheme of drug loading onto the hyaluronic acid/spider silk-based hydrogels. The drug loading capacity and the entrapment efficiency of the hyaluronic acid/spider silk-based hydrogels were measured using the equations described in the experimental section. Entrapment efficiency was measured compared to the initial drug concentration that was applied for the diffusion induced hydrogel drug loading. The entrapment efficiency of the hyaluronic acid/spider silk-based hydrogels ranged between 48–62% of the initial drug concentration, which demonstrated fluctuations of the entrapment efficiency compared to the different component ratios of hyaluronic acid and spider silk. Increased hyaluronic acid component ratios in the hyaluronic acid/spider silk-based hydrogel preparations show the increment of entrapment efficiency, suggesting that the hyaluronic acid in the hyaluronic acid/spider silk-based hydrogels had directly affected the entrapment efficiency. Increased preparatory ratios of hyaluronic acid can increase the extensibility of the hydrogel polymer, which can acquire relatively high amounts of substances by diffusion. The effects of spider silk proportions in the hyaluronic acid/spider silk-based hydrogels indicate a slight decrement of ibuprofen entrapment efficiency. These results show that the effects of the spider silk component in hyaluronic acid/spider silk-based hydrogels are directly less considerable in drug entrapment evaluations of hydrogels. An increased spider silk proportion in the hyaluronic acid/spider silk-based hydrogels leads to a tenderer crosslinking with a higher crosslinking density. The high crosslinking density and tender covalent crosslinking have slight effects on the extensibility of the hyaluronic acid/spider silk-based hydrogels, which is inversely proportional to the entrapment or loading of the substances within the hydrogel polymeric structure.

The drug loading capacity was obtained using the maximum drug amount that hyaluronic acid/spider silk-based hydrogels could hold per unit by weight. In these experiments, the known concentration of ibuprofen volume introduced into the hyaluronic acid/spider silk-based hydrogels and the maximum drug uptake by unit volume of hydrogels were measured. Hyaluronic acid/spider silk-based hydrogels show extraordinary drug loading amounts, as shown in Figure 4c. Increased hyaluronic acid proportions of the hydrogel preparations show comparatively increased drug loading ratio, with the maximum drug loading capacity of 4.88% (*W*/*W*) from the total ingested weight of hydrogels. The spider silk proportions of the hydrogels show a slight decline in the drug loading capacity of the hyaluronic acid/spider silk-based hydrogels. Hydrogels can absorb the anti-inflammatory drug ibuprofen in the range of 3.98–4.88%, suggesting hyaluronic acid/spider silk-based hydrogels are optimal for use in external drug delivery systems as a local anti-inflammatory drug carrier.

### 3.10. Drug Release of Hyaluronic Acid/Spider Silk Hydrogels

The drug release studies of the hyaluronic acid/spider silk-based ibuprofen loaded hydrogels were observed as described in drug release experiments. Ibuprofen release from the hydrogel structures was facilitated by concentration gradient diffusion of drug molecules under physiological conditions (PBS pH 7.2, 37 °C). (Figure 4b) The external chamber of the drug release medium was refreshed during predefined time intervals, leading to the overcoming of the effects on the diffusion by drug accumulation in the release medium. Subsequently, the refreshment of PBS medium at defined time intervals facilitated the illustration of drug release content at corresponding time intervals. UV-Vis spectroscopy was used to describe the ibuprofen release concentrations corresponding to the predesigned calibration curve for ibuprofen that was found to present clear absorption peaks at 221 nm. Drug release studies of hyaluronic acid/spider silk-based drug loaded hydrogels show an excellent ability to release ibuprofen with the approximate release percentage of 80% from the initial drug content entrapped within the hyaluronic acid/spider silk-based hydrogels (Figure 4d). Effects of the different component ratios of hyaluronic acid/spider silk-based hydrogels are illustrated in the supporting information. Even though the effects of the component ratios on the drug delivery do not show extreme deviations, slight effects of the component ratios on the drug delivery were observed. (Appendix A) These results suggest that the Ibuprofen release profiles of hyaluronic acid/spider silk-based hydrogels are optimum to use in anti-inflammatory drug delivery applications.

### 3.11. Conductivity of Hyaluronic Acid/Spider Silk Hydrogels

The conductivity of the hydrogels is at the lower margin of the conductivity of semi conductive materials (5–6 × 10^−6^ S/m). HA/Ss 1 (Table 1). Hydrogels show the lowest conductivity while HA/Ss 3 (Table 1) hydrogels exhibit the highest conductivity from the prepared sample concentrations. According to the different component ratios in the hydrogel preparations, the conductivity variation was illustrated in the supporting information. The conductivity and relevant resistance of the HA/Ss 2 hydrogel are shown in Figure 4e, which was earlier described as the representative preparation for considering the characteristics and properties of the HA/Ss hydrogels as a whole. An increase in the conductivity of hyaluronic acid/spider silk-based hydrogels was observed with an increase in the hyaluronic acid component ratio within the studied composition range. An increase in the spider silk component ratio of hydrogels demonstrates a slight but noticeable increase in conductivity. The conductivity results of the hydrogels suggest the applicability of hyaluronic acid/spider silk-based hydrogels in electrophoretic drug delivery systems where the drug-loaded hydrogel medium is dielectric, allowing for the delivery of the drug while specifically preventing the delivery of other elements that occupy the hydrogels [92,93].

### 3.12. Enzymatic Degradability of Hyaluronic Acid/Spider Silk Hydrogels

Enzymatic degradation of hyaluronic acid/spider silk-based hydrogels shows different degradation levels when exposed to enzyme trypsin and enzyme chymosin. Trypsin and chymosin are digestive tract proteolytic enzymes that degrade proteins and polymeric structures into their respective oligomers or monomers (Figure 5a). Trypsin is the most commonly found proteolytic enzyme in most of the vertebrate digestive tract (particularly the duodenum), and is a member of PA clan superfamily of proteases [94,95]. Trypsin is mainly produced in the form of trypsinogen by the acinar cells in the pancreas [95]. Trypsin is considered to be one of the major proteolytic digestive enzymes in the human digestive system. Chymosin (also referred to as Rennin) is mainly produced in the ruminant stomach, and is one of the main proteolytic initiation enzymes in the young ruminant digestive tract [96]. Chymosin is produced in the chief cells in the stomach, and initiates the breakdown or coagulation of milk proteins. Chymosin is considered an aspartic protease [97]. Trypsin degradability shows more degradation of the hydrogels compared to chymosin. The degradability in the face of both trypsin and chymosin was increased for samples containing less hyaluronic acid. In addition, the samples with increased spider silk ratios with the same amount of hyaluronic acid show less degradability when exposed to chymosin and trypsin (Appendix A). Looking at the data, we can see 15–35% degradability of hyaluronic acid/spider silk-based hydrogels with chymosin. With trypsin, the degradability of hyaluronic acid/spider silk-based hydrogels ranges between 25–50% of the initial mass (Figure 5b,c). Analyzing the results, the degradability of hyaluronic acid/spider silk hydrogels is partially degraded by proteolytic enzymes. Increasing the spider silk component decreases the degradability, while increasing the hyaluronic acid component increases degradability in the presence of chymosin and trypsin. The enzymatic degradability behavior suggests that the partially retarded proteolytic capabilities of spider silk have affected the degradability, which decreased with an increased in spider silk composition. The enzymatic degradability of hyaluronic acid/spider silk-based hydrogel is shown in Figure 5a.

### 3.13. Cell Viability Studies of Hyaluronic Acid/Spider Silk Hydrogels

The hyaluronic acid/spider silk hydrogels are assumed to be widely applicable in the biomedical field. Considering this assumption, a cytotoxicity assay (MTT assay) was performed in order to identify the impact of the hydrogels on biomedical parameters. MTT assay is a colorimetric test that estimates the metabolic activity of cells. Human postnatal fibroblast (HPF) cells were used for the cell cytotoxicity experiments as a model cell culture that synthesizes an extracellular matrix [98,99]. 

The survival rate was determined for hyaluronic acid/spider silk-based hydrogel (max. concentration 15 mg/mL of hyaluronic acid and 3 mg/mL of spider silk; min concentration 0.12 mg/mL of hyaluronic acid and 0.025 mg/mL of spider silk) (Figure 5f) and its components (HA (0.06 mg/mL:15 mg/mL) (Figure 5e), spider silk (0.0125 mg/mL:3.2 mg/mL) (Figure 5f)) by MTT assay for 72 h. Spider silk by itself showed no cytotoxicity (87% cell viability), but HA alone demonstrated a little cytotoxicity at the highest concentration tested (67% cell viability). Nevertheless, it was observed that the tested hyaluronic acid/spider silk-based hydrogel had no significant influence on cell viability (84%) at the maximal concentration based on a control with an equal amount of PBS solution. It demonstrates that spider silk somehow influences hyaluronic acid’s ability to reduce cell metabolic activity. The decrease in cell viability for hyaluronic acid hydrogels was also observed by parallel studies with hyaluronic acid above 1 mg/mL of concentration on different cell lines [100,101]. Moreover, it is important to note that spider silk showed the ability to neutralize the negative effects of hyaluronic acid.

The seeded human postnatal fibroblast cells were observed under a light microscope under the magnifications of 10×, 20×, and 40×. Two-dimensional cell growth and proliferation were observed over 72 h, with observations every 24 h. HPF cells were successfully cultured within the 72 h period, as shown in Figure 5g. Accordingly, the cell morphology or cellular structures were not deformed, and the cells showed normal morphology over the 72 h duration. The results suggest that the hyaluronic acid/spider silk-based hydrogels are not cytotoxic for external use (skin contact). According to the received data, the studied concentrations of hyaluronic acid/spider silk hydrogels have potential uses in further biomedical and biological applications.

### 3.14. Antimicrobial Ability of Hyaluronic Acid/Spider Silk Hydrogels

Hyaluronic acid/spider silk-based hydrogels show excellent antimicrobial ability on both gram-positive and gram-negative bacteria. Antimicrobial experiments were conducted by inhibitory zone experiments on both types of bacteria (*Micrococcus sulfuricum* and *Escherichia coli*). The diameters of the inhibitory zones were measured via eight cross-sections of the inhibitory bands, and the average inhibitory range demonstrates the relative antimicrobial ability of the hyaluronic acid/spider silk-based hydrogels. Antimicrobial ability increased with an increase in the hyaluronic acid component ratio for both *Micrococcus sulfuricum* and *Escherichia coli* bacteria (Figure 6c). An increase in the spider silk component ratio in the hydrogel preparations, meanwhile, caused a slight increase in the inhibitory range surrounding the hydrogel samples with both *Micrococcus sulfuricum* and *Escherichia coli* (Figure 6c). Comparative analysis of the gram-positive and gram-negative bacterial inhibitory ranges suggests that the hyaluronic acid/spider silk-based hydrogels have higher antimicrobial ability with gram-positive bacteria (*Micrococcus sulfuricum)* than gram-negative bacteria (*Escherichia coli*) (Appendix A). The ultimate results of the antimicrobial tests demonstrate exceptional antimicrobial ability, which suggest the applicability of hyaluronic acid/spider silk-based hydrogels as a magnificent antimicrobial biomaterial in a variety of biomedical applications. The origins of the antimicrobial ability of hyaluronic acid/spider silk-based hydrogels lie in both the hyaluronic acid and the spider silk components. Hyaluronic acid possesses inhibitory effects on bacterial attachment with bacteriostatic properties, while spider silk (*Linothele fallax*) shows inhibitory effects on bacterial adhesion. Hyaluronic acid interferes with bacterial growth by inhibiting protein synthesis and bacterial replication [65,102]. Hyaluronic acid has demonstrated concentration and molecular weight-dependent antimicrobial ability on both gram-positive and gram-negative bacteria (10–20 mg/mL) (Appendix A). The antimicrobial rate of the hydrogels is represented by the length of bacteria inhibition zones according to the different component ratios (Figure 6c,d (Appendix A)). Bacterial growth on the hydrogel surface shows a lack of bacterial attachment compared to the control sample with agar. There were no such profound colonies on hydrogel surfaces, with the exception of some isolated colonies. The results suggest that the combination of hyaluronic acid and spider silk inhibits the bacterial attachment with their bacteriostatic and anti-adhesive properties. The illustrations and experimental results of the antimicrobial ability of hyaluronic acid/ spider silk-based hydrogels are shown in Figure 6a–c.

## 4. Conclusions

Hyaluronic acid/spider silk-based hydrogels were successfully prepared using EDC/NHS crosslinking, which facilitates the cross-coupling between carboxylic acid and amine functional groups. Results demonstrate that hyaluronic acid/spider silk-based hydrogels are capable of a wide range of biomedical applications. Scanning Electron Microscopic images and Fourier-transform infrared spectroscopy demonstrated the fibrillar crosslinked structures and the exact compositions of the hydrogels, respectively. Contact angle measurements show the hyaluronic acid/spider silk-based hydrogels have extraordinary wettability, which confirms their hydrophilic characteristics. Their swelling degrees demonstrate that hyaluronic acid/spider silk-based hydrogels are capable of acquiring and holding 15–30 times their dry weight in water (pH = 7.2, 25 °C) within their porous structures. The swelling degree suggests hyaluronic acid/spider silk-based hydrogels have extraordinary potential to entrap liquefied drug molecules within their porous structures, which could lead to their use in diffusion-induced drug delivery. The shrinking abilities of the hyaluronic acid/spider silk-based hydrogels under physiological temperature (37 °C) show that the hydrogels can provide external support for drug release under physiological conditions by shrinking their crosslinking polymeric scaffold. Viscosity measurements of hyaluronic acid/spider silk-based hydrogels show shear-thinning properties, which suggest the hydrogels are a promising material for 3D printing. The conductivity of the hyaluronic acid/spider silk-based hydrogels is optimum for electrophoretic drug delivery due to their dielectric properties. Considering the cytotoxicity of the hydrogels, human postnatal fibroblasts (HPF) cells were overgrown without morphological changes, confirming their biocompatibility. As the findings of the drug loading and entrapment efficiency show, hyaluronic acid/spider silk-based hydrogels can carry an effective and optimum amount of ibuprofen for local anti-inflammatory treatments. Drug release of hyaluronic acid/spider silk-based hydrogels presented an ibuprofen release of more than 80% compared to its entrapped drug amount.

Concerning the ultimate and the spotlighting aspect of the research, hyaluronic acid/spider silk-based hydrogels show excellent antimicrobial ability on gram-positive and gram-negative bacteria, which suggests a vast range of antimicrobial applications such as wound dressings, disinfection coatings, catheter coatings, prosthetic implant coatings, storage media for surgical and clinical instruments, and 3D printing of antimicrobial surgical accessories. Ibuprofen-loaded hyaluronic acid/spider silk-based hydrogels facilitate multiple antimicrobial sources, which can delay or diminish the occurrence of antimicrobial drug-resistance as an alternative antimicrobial and anti-inflammatory agent for external use.

## Figures and Tables

**Figure 1 polymers-13-01796-f001:**
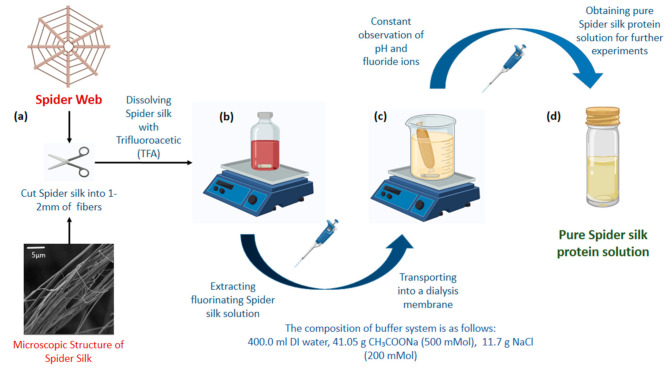
Scheme of the spider silk protein solution preparation. (**a**) Obtaining and cutting spider silk into 1–2 mm pieces (reduces dissolution time). (**b**) Fluorinated spider silk solution after constant mixing with trifluoroacetic acid (TFA). (**c**) Dialysis of fluorinated spider silk solution against buffer (400 mL DI water, 41.05 g CH_3_COONa (500 mMol), 11.7 g NaCl (200 mMol)). (**d**) Obtaining pure spider silk solution after constant observations of pH and fluorine ions with CaCl_2_ quality reactions.

**Figure 2 polymers-13-01796-f002:**
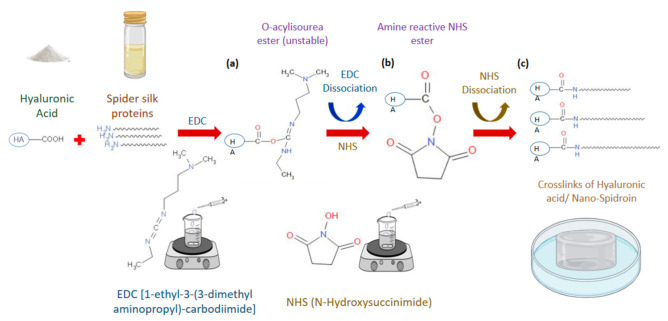
Schematic mechanism of the synthesis of hyaluronic acid/ spider silk-based hydrogels by EDC (1-ethyl-3-(3-dimethyl aminopropyl)-carbodiimide/ NHS (N-Hydroxysuccinimide) crosslinking (pH = 4.5–5.5). (**a**) EDC reacts with the carboxylic groups to produce unstable O-acylisourea intermediate. (**b**) NHS directly dissociates EDC, leading to the formation of amine reactive NHS ester. (**c**) The highly reactive NHS ester directly binds with amine groups in the spider silk, leading to the production of a crosslinked hydrogel structure.

**Figure 3 polymers-13-01796-f003:**
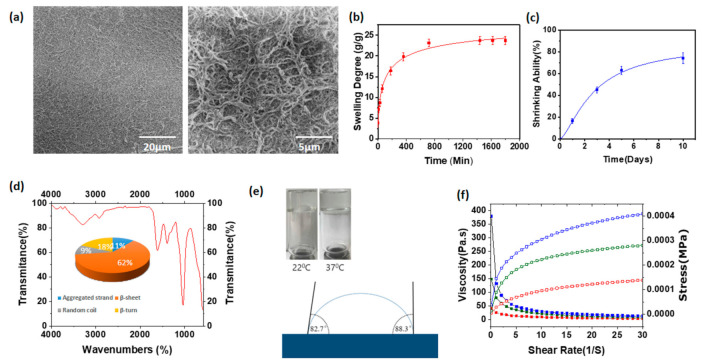
(**a**) Scanning electron microscopy images of hyaluronic acid/spider silk-based hydrogels under magnifications of 20 µm and 5 µm. (**b**) Swelling degree of hyaluronic acid/spider silk-based hydrogels. (**c**) Shrinking percentages of the hydrogels compared to initial weight over a 10 day period. (**d**) FTIR spectroscopic analysis and secondary structural analysis of hyaluronic acid/spider silk-based hydrogel. (**e**) Contact angle measurements of hyaluronic acid/spider silk-based hydrogel by KRUSS and flip test of hydrogels at 22 °C and 37 °C. (**f**) Viscosity and rheological properties of HA/Ss hydrogel at pH 7.2, 22 °C; pH 7.2, 37 °C; and pH 4.5, 22 °C conditions.

**Figure 4 polymers-13-01796-f004:**
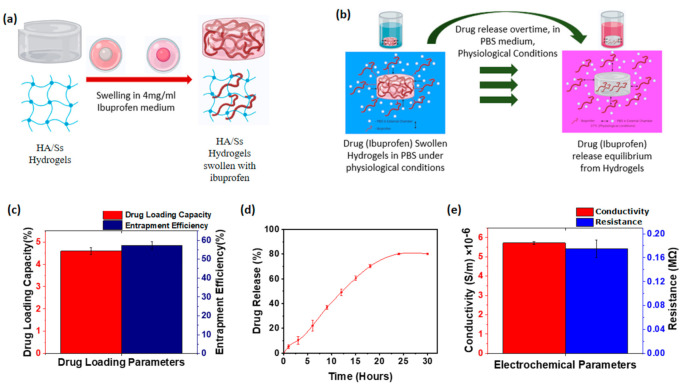
(**a**) Schematic diagram of drug (ibuprofen) loading on hyaluronic acid/spider silk-based hydrogels. (**b**) Schematic diagram of drug (ibuprofen) release from hyaluronic acid/spider silk-based hydrogels. (**c**) Drug (ibuprofen) loading capacity and drug (ibuprofen) entrapment efficiency of hyaluronic acid/spider silk-based hydrogels as drug (ibuprofen) loading parameters (HA/Ss 2). (**d**) Drug (ibuprofen) release (%) over time (HA/Ss 2). (**e**) Resistivity and conductivity of the hydrogels representing electrochemical parameters of the hyaluronic acid/spider silk-based hydrogels (HA/Ss 2).

**Figure 5 polymers-13-01796-f005:**
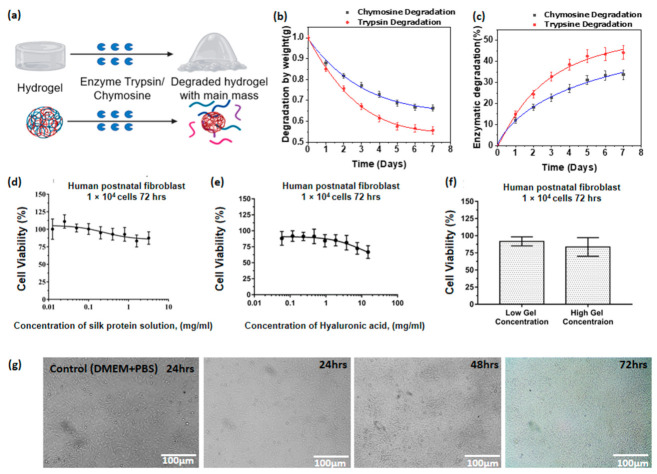
(**a**) Degradation process of hydrogels with enzyme chymosin and enzyme trypsin under physiological conditions (37 °C), degraded minor polymer particles surrounded by main mass are shown. (**b**) Degradation by weight of hyaluronic acid/spider silk-based hydrogels with enzyme chymosin and trypsin over seven days (HA/Ss 2). (**c**) Degradation percentage of hyaluronic acid/spider silk-based hydrogels with enzyme chymosin and trypsin compared to their initial weight (HA/Ss 2). (**d**) Cytotoxicity assay (MTT assay) of spider silk solution according to the hydrogel preparatory concentrations. (**e**) Cytotoxicity assay (MTT assay) of hyaluronic acid solution according to the hydrogel preparatory concentrations. (**f**) Cytotoxicity assay (MTT assay) of hyaluronic acid /spider silk mixtures according to the hydrogel preparatory concentrations. High gel concentration (HA concentration = 15 mg/mL; spider silk concentration = 3 mg/mL); low gel concentration (HA concentration = 0.12 mg/mL; spider silk concentration = 0.025 mg/mL). (**g**) Cell viability on hyaluronic acid/spider silk-based hydrogels (HA/Ss 2) with human fibroblast cells after 24 h, 48 h, and 72 h of cell proliferation intervals in comparison with the control (DMEM + PBS).

**Figure 6 polymers-13-01796-f006:**
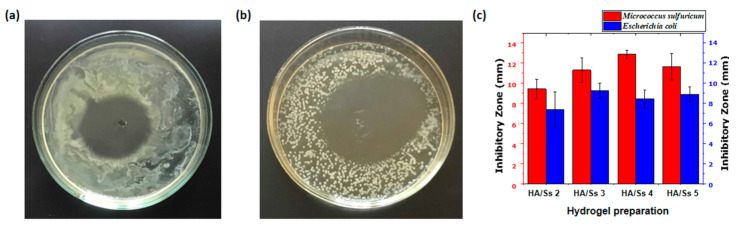
(**a**) Inhibitory band test of hyaluronic acid/spider silk-based hydrogels on gram negative *Escherichia coli* bacteria. (**b**) Inhibitory band test of hyaluronic acid/spider silk-based hydrogels on gram positive *Micrococcus sulfuricum* bacteria. (**c**) Inhibitory zone length of hyaluronic acid/spider silk-based hydrogels on gram negative (*Escherichia coli*) and gram positive (*Micrococcus sulfuricum*) bacteria.

**Table 1 polymers-13-01796-t001:** Component ratios of hyaluronic acid/spider silk (HA/Ss)-based hydrogels.

Sample	Spider Silk	Hyaluruonic ACID	EDC/NHS	MES Buffer (0.1 mol/L)
HA/Ss 1	18 mg	100 mg	1 ml	8 mL
HA/Ss 2	18 mg	150 mg	1 ml	8 mL
HA/Ss 3	18 mg	200 mg	1 ml	8 mL
HA/Ss 4	22.5 mg	150 mg	1 ml	7.5 mL
HA/Ss 5	27 mg	150 mg	1 ml	7.5 mL

**Table 2 polymers-13-01796-t002:** Gel fraction and crosslinking density of hydrogels.

No.	Hydrogel Preparation	Gel Fraction (%)	Crosslinking Density ×10^−4^ mol/cm^−3^
01	HA/Ss 1	79.83	7.27
02	HA/Ss 2	79.86	8.11
03	HA/Ss 3	80.09	7.81
04	HA/Ss 4	81.91	8.20
05	HA/Ss 5	83.86	8.43

## Data Availability

The data presented in this study are available on request from the corresponding author.

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
