# Peer review of "Native Spider Silk-Based Antimicrobial Hydrogels for Biomedical Applications"

_polymers, 2021, doi:10.3390/polym13111796_

Round 1

Reviewer 1 Report

This manuscript reported the preparation and biomedical properties of hydrogels based on hyaluronic acid (HA) and spider silk (Ss) through chemical crosslinking approach. Definitely, the authors studied the swelling, rheological, antimicrobial, and related biomedical properties carefully, particularly the antibacterial performance is interesting for application in wound dressings. But the follow questions should be mentioned and the manuscript needs a further revision.

i)In 3. Results and Discussion, the authors can list sub-titles to state the major results clearly. For example, 3.1 Characterization of chemical structure; 3.2 Morphology and swelling ability, ----.

ii)In Figs.3(b)(c)(f), the digitals are too small, the readers are difficult to distinguish them.

iii)The antimicrobial property is a major content in this manuscript, the related results in supporting information can be added to the text. In addition, it is necessary to supplement the quantitative results on antibacterial performance, finally the antibacterial rate is available.

iv) Antibacterial mechanism needs further analysis.

v) Last part, Conclusion should be written as 4. Conclusions with the same format as the former parts.

It can be accepted for publication after a revision.

Author Response

Dear reviewer,

We would like to take this opportunity to thank all the reviewer for the valuable advices and insightful comments. The manuscript was corrected following comments from the reviewers. Visual modifications and language modifications were made concerning valuable suggestions by reviewer. Authors have revised the modified manuscript, and sincerely hope that each concern was satisfactorily addressed in the revised manuscript.
Truly,
Elena F. Krivoshapkina.

Reviewer 2 Report

The paper describes the obtaining of novel antimicrobial natural polymeric hybrid hydrogels based on hyaluronic acid and spider silk by chemical crosslinking method.

The results are proper for publishing in this journal, however, there are issues that must be solved before its recommendation for publication, such as:

- Please, check the english of the manuscript and repetitive words. Extensive editing of English language and style are required. E.g. „Hydrogel polymers are synthetic and also can be natural polymers”; „The porosity, soft consistency, and ability to absorb biological fluids increased the ability to use hydrogels in biomedical applications.” ; -Table 1. Hyaluruonic Acid

-  Please, add the importance of your study and originality. If there are stusied HA-spider silk protein, the authors should mention what this study brought new.

- The obtained hydrogels have been washed after synthesis? The residual monomer was removed? The gel fraction should be calculated and the crosslinking density, too. How the introduction of spider silk affected these parameters?

- Please revise the figures from the manuscript. Font of the axes and also the legends should be magnified, even these were presented at Supp.Info.

-Conclusion should be presented more concisely.

Therefore I would suggest publication of the paper after the major revisions are taken into consideration.

With respect,

Author Response

(The authors gave the same response as above.)

Round 2

Reviewer 2 Report

Dear Authors, 

The manuscript has been improved and can be published.

With respect,